# An approach to increase the power output of Karnafuli Hydroelectric Power Station: A step to sustainable development in Bangladesh's energy sector

Fakir Sharif Hossain[1]☯*, Tahmidur Rahman[1]☯, Abdullah Al Mamun[1]☯, Omar Bin Mannan[1]☯, M. Altaf-Ul-Amin[2]

**1** Department of Electrical and Electronic Engineering, Bangladesh Army International University of Science and Technology (BAIUST), Cumilla, BD, **2** School of Information Science, Nara Institute of Science and Technology (NAIST), Ikoma, Nara, Japan

☯ These authors contributed equally to this work.
* dr.hossain@baiust.edu.bd

**Data Availability Statement:** All relevant data are within the paper.

## Abstract

Renewable energy has become the most prominent source of energy to reduce carbon emissions around the globe. Undoubtedly, hydro energy is very much clean energy among other sources. In Bangladesh, hydro energy is available only in a specific southern area contributing several hundred megawatts to the national grid. This paper devotes to assessing the capacity and practicability of a hydropower plant to boost the power output by implementing the combined cycle hydropower system. The proposed method has been developed by 1) studying the existing plant based on surveyed data, 2) selecting the site for installing the hydrokinetic turbine, 3) designing with consideration of numerous constraints of inter dependability, and 4) creating a prototype model to ensure the practicability. Preliminary results show that a significant amount of additional electric energy can be generated from the plant with higher efficiency.

## 1 Introduction

Sustainable Development Goal (SDG) 7 set by United Nations (UN) asks to ensure access to electricity to the citizens of the poorer countries with the source of clean energy in an affordable cost [1]. This goal has several sections with different meanings and implications. The UN has wholly articulated the focused areas to improve the achievability of the aim targeting two areas. The first target of SDG7.a imposes on researching clean energy along with fossil fuel. The second target area of SDG7.b focuses on upgrading and expanding sustainable energy and infrastructure for developing countries, particularly for least developed and landlocked countries. There is a growing list of disadvantages of using coal as an energy source in power plants. People living near the mines are always at the risk of displacement due to total environmental

**Funding:** NO-The author(s) received no specific funding for this work.

**Competing interests:** The authors have declared that no competing interests exist.

destruction [2–4]. As one of the worst climate change victims globally, should not Bangladesh (BD) set examples by reducing its emission?

To be a part of the mission of SDG 7, a least developing country like BD has numerous potentiality in renewable and clean energy resources [5]. In BD, the current population is 163 million, but not all of them have the chance of consuming electricity [6]. The government of BD is investing a lot in gas and coal-powered power plants to meet the energy demand. However, the effort must still be magnified by searching for some alternative renewable sources for energy generation. There are many potential resources, such as solar, wind, hydro, etc., in BD, wherein hydropower supports approximately one and a half percent of the total electricity generated in BD [7]. This small percentage of hydropower is generated from the only hydro plant named Karnafuli hydroelectric power Station (KH), located near the port city of Chittagong [8].

Hydro energy can be classified into two types namely, hydrostatic and hydrokinetic. On the one hand, hydrostatic energy depends on the head difference between the water surface and the turbine maintained by constructing dams and water reservoirs. Though there are numerous environmental threats to fish migration paths, wildlife etc. in creating those sorts of barriers, most of the traditional hydro plants are using the hydrostatic technique to extract energy [9–11]. On the other hand, the horizontal velocity of water current is the main actuating factor for driving turbines to generate hydrokinetic power. This kinetic energy is more environmentally friendly as it does not require dams or water Reservoirs [12]. The hydrostatic energy conversion method is currently applied for energy generation in the KH plant. Potentially, hydrokinetic energy is another energy source that can generate some extra power from the tailwaters in the KH dam. The water that is discharged to the river as tailwater contains enormous hydrokinetic energy and has a great potential to be converted into electric power utilizing the existing infrastructure of KH [13]. At present, this vast hydrokinetic energy goes wasted in the KH plant. This paper proposes a scheme to utilize the wasted energy by implementing a combined cycle hydropower system (CCHS) in the existing KH plant.

## 1.1 Related works and motivation

Traditional hydropower plants usually harness the hydrostatic potential to generate electricity. A few works have been in the literature based on hydrokinetic energy, and it is considered a new scope of research [14–17]. Hydrokinetic power generation has a great potentiality and still requires further research on various aspects: data integrity, operation and maintenance, shutdown arrangements and security, starting mechanism, depth of knowledge, and cost analysis [18–22]. However, there are some studies in the literature to display the implementation cost of hydrokinetic turbines compared to hydrostatic counterparts such as the unavailability of different components, and independence of design optimization [21, 23–24]. In [24], for example, Savonius type hydrokinetic turbine efficiency has been studied. They present a review of different parameters affecting the performance of the Savonius hydrokinetic turbine, and later on, upgrade the design in [21]. A practical hydrokinetic turbine's output was 300kW which was established in 2003 by Marine Current Turbines Limited, and later on, a larger capacity turbine of 1MW power was installed in Korea in 2009 [25]. Unlike wind, and solar energy plants, the hydrokinetic turbine cannot extract a significant amount of energy due to inefficient design and other factors [26, 27]. It is, therefore, required to design optimized hydrokinetic turbines for extracting more energy from hydrokinetic resources.

The CCHS method utilizes both types of energy for generating power. In the existing CCHS model, all works are based on a river water current where there is a shallow head [28–34]. Hasting HPS in the USA in the Mississippi river is installed to generate hydrokinetic

power by using two modes of turbines in series in the tailwaters of the project [28]. In [29], a case study is presented based on the Tucurui hydroelectric power station in Brazil. They estimate the hydrokinetic energy potential at the outflow of a hydropower plant. However, the methodology depends on the river stream velocity and depth compatible with the turbine diameter. In [30], authors estimate hydrokinetic potential in two Brazilian rivers. The methodology allows studying maximum velocities and turbine arrangement, and estimated electrical power in both rivers was equal to 109.5 and 31.5 KW. The method heavily depends on simulation and river current velocity. In [31, 32], Hydrokinetic resources under the influence of tide and river discharges are investigated in the Amazon river. They show that the river discharges lead to a considerable increase in performance during winter. These methods utilize the seasonal water current in the river. A methodology is presented in [33] to assess the hydrokinetic potential of the Eastern Yamuna central canal. Though the study provides a maximum potential of 26.48 MW is possible, however, it depends on the river head and water current frequency. In [34], authors show that a portion of the Brazilian Amazon river with real measured data on velocity, curvature, and geometry in the field can deliver potentiality. Results are convincing; however, the total construction cost is very high on the river bank.

Another research work in [35] shows a theoretical analysis and demonstrates that the remaining velocity after the hydrostatic turbine in the draft tube is waste, but utilization can be possible for generating energy by hydrokinetic turbines. A similar claim can be seen in [18] where authors conduct installing a horizontal turbine in series of the traditional vertical turbine in a river dam. Very few works can be identified that contributes in designing the CCHS technology [22, 36–38]. In [22], the authors discuss a potential study of applying hydrokinetic turbines in the tailwaters of existing conventional hydropower stations. This paper opens a gateway for researching CCHS technology. Based on this work, some other research works are published [36, 37]. In [36], authors implement floating propeller-shaped hydrokinetic turbines in the outflow channel of a hydropower plant. They monitor the system through wireless sensor networks and the Internet of Things (IoT). The system can not capture the whole water current velocity due to the floating concept. In [37], a method is applied to the downstream reservoir of the Tucuruí hydroelectric dam. They can generate 3 MW power based on simulation by placing 73 turbines. The region of interest is a reservoir to turbine unit. In Indonesia, [38], a survey on hydrokinetic potential in Balambano hydropower PT Vale is carried out to demonstrate the feasibility of placing hydrokinetic turbines in the tailrace. The work displays the possible site selection based on the water velocity in the outflow of the dam. However, the method does not analyze the dependency on the existing plant.

Therefore, there is a scope of getting augmented output power from an existing hydropower plant through the CCHS model. This work proposes of installing a hydrokinetic turbine in series with the existing hydrostatic turbine in the KH plant to generate significant amount of extra power. There is a scope of designing a hydrokinetic turbine to use the horizontal water current in an optimum way to increase the plant's output. Several technical challenges are required to meet for obtaining augmented power output. Site selection, turbine installation, turbine design, and analysis of numerous constraints of inter dependability are the significant considerations to be addressed.

The proposed method aims to obtain additional power from the existing plant utilizing the CCHS technology to increase overall plant capacity. The main contributions of this paper are summarized as follows.

- Surveying in Kaptai hydropower plant to collect data. We collect the actual data of the KH plant.

- Primary analysis and power estimation are performed based on surveyed data. A simulation model is established to analyze the existing KH plant.

- Selecting the region of interest to install the hydrokinetic turbine.

- A CCHS technology is proposed to design a hydrokinetic turbine to augment the plant's power output in parallel to the existing setup. The result of the existing simulated model and proposed design is compared to verify the feasibility of the extra setup.

- Implementation of a prototype model on a smaller scale and comparison with the simulated results.

The remaining section of this paper is organized as follows. Section 2 describes the present energy scenario of Bangladesh and how far to go to achieve SDG7. Karnafuli Hydroelectric Power Station and specifications of its various equipment are discussed in section 3. Historical data and the existing plant simulation model of the study area are presented in section 4. In section 5, the proposed method is explained. Finally, it concludes the work by showing results in section 6.

## 2 Preliminaries and background

### 2.1 SDG7 Vs. low middle-income country Bangladesh

Goal 7 suggests ensuring the right of all people to use cheap, renewable, and secured modern energy resources. In the context of the social, geo-environmental perspective of Bangladesh and the people of Bangladesh, some dimensions of achievability are required to reflect on attaining the sustainable development goal-7 [39].

Energy is considered one of the basic needs of countries with poor sustainability regarding natural resources such as surface water availability, sustainable creativity, soil fertility, seasonality, and biodiversity. Energy specially electricity is generally considered as the source of socio-economic prosperity. On the one hand, there is a shortage of energy resources in BD such as coal, oil, gas, and uranium. On the other hand, energy management in BD also does not come within the framework of ensuring access to energy for all [40]. Renewable energy is an alternative energy source that can harness natural resources like biomass, sunlight, rivers, waterfalls, ocean waves, wind, tides, etc. Goal 7 will remain unattainable unless Bangladesh can develop sustainable energy generation technologies.

Obviously, in villages in the mainland, household rooftop solar PV and wind turbines, respectively requiring uninterrupted sunlight and wind flow are impractical. Standing on the verge of the technology-intensive fourth industrial revolution (Klaus Schwab, 2016), Bangladesh can hope to achieve SDGs. The nation can recreate prosperous human society and ecosystem-based energy in harmony with its natural resources and environment to achieve sustainable development with an emissions-neutral self-reliant life. While the West needs to address energy-related decarbonization as a matter of urgency, countries like Bangladesh should hold to their carbon-neutral living culture and heritage to ensure future sustainability [40].

### 2.2 Energy scenario of Bangladesh

Bangladesh is working positively to increase electricity generation. A significant percentage of about 94% has access to electricity with a maximum generation of 22,562MW [1]. However, this sector is still facing challenges to ensure reliable power supply due to natural gas shortage, aging of plant equipment and insufficient transmission and distribution networks. In addition, the projected electricity demand by 2030 is approximately 27.4GW. More spotlights are required to put on renewable energy to meet the demand. However, the electricity generation

by renewable energy sources is around 3% of total energy generation in BD [5]. A considerable percentage of renewable energy is generated through solar and a little from hydropower.

## 2.3 Prerequisites of hydro power plant

Numerous components are essentials to convert hydro energy to electrical power [41]. Although a massive list of equipment and infrastructure is required to facilitate a smooth power generation, here we discuss some of them. The top focusing points are reservoir, dam, penstock, spillway, draft tube, tailwaters, turbine, and generator. Water reservoir stores water in rainy seasons and delivers water to the turbine as input. There are dams and penstock between the water turbine and reservoir to maintain a good quality of water static energy. The dam provides a head, and the penstock carries the water to the turbine through a channel. Any fluctuation in water flow is handled by forebay and delivers a linear water flow to the surge tank. The surge tank diminishes the pounding effect and creates pressure in the penstock. The spillway reduces excessive water in the reservoir during the flood.

In the electricity generation part, there is a powerhouse to support and secure the electric equipment. Generators are essential components in the powerhouse. Water turbines work as the prime mover for generators or alternators. Water turbines require being highly efficient and robust in construction to peak up a load in a brief period. They are built in all sizes up to approximately one-tenth of a million horsepower with speed varying from 100 r pm (revolution per minute) to 1000 r pm based on turbine size. The alternator converts the mechanical energy from prime mover to electric power with the help of electromagnetic field induction.

## 3 Description of study area: Karnafuli Hydroelectric Power Station (KH)

The study area, site selection, data collection, data analysis, and proposed turbine model are presented with mathematical analysis and simulation. The following notations are used to explain the proposed method.

- SDG: Sustainable development goal.

- $P$: Total power generated from the hydroelectric turbine of area $A$ with water velocity $V$.

- $KVY$ and $KV\Delta$: Kilovolt for star and delta connected transformer or generator.

- PLL: Programmable logic array is used for the simulation of unit-4 of the KH plant in matlab simulink.

- $P_m$: Mechanical power input to the generator from the turbine unit.

- $V_a$: Generated voltage.

- $PWM$: Pulse width modulator is used for the simulation of unit-4 of the KH plant in matlab Simulink.

- PMDC generator: Permanent magnet direct current (DC) generator.

- $V_{tidal}$ and $V_{after_t}$: are the tidal and water velocity (after the turbine) respectively.

- $F_W$ and $P_{turbine}$: Water force and turbine output power.

- $V_{Wcurrent}$, $Q_{turbine}$ and $A_{ch}$: these parameters are water current velocity, turbine discharge quantity, and exit area of a channel of the draft tube.

- $P_{out}$: Hydrokinetic water current power.

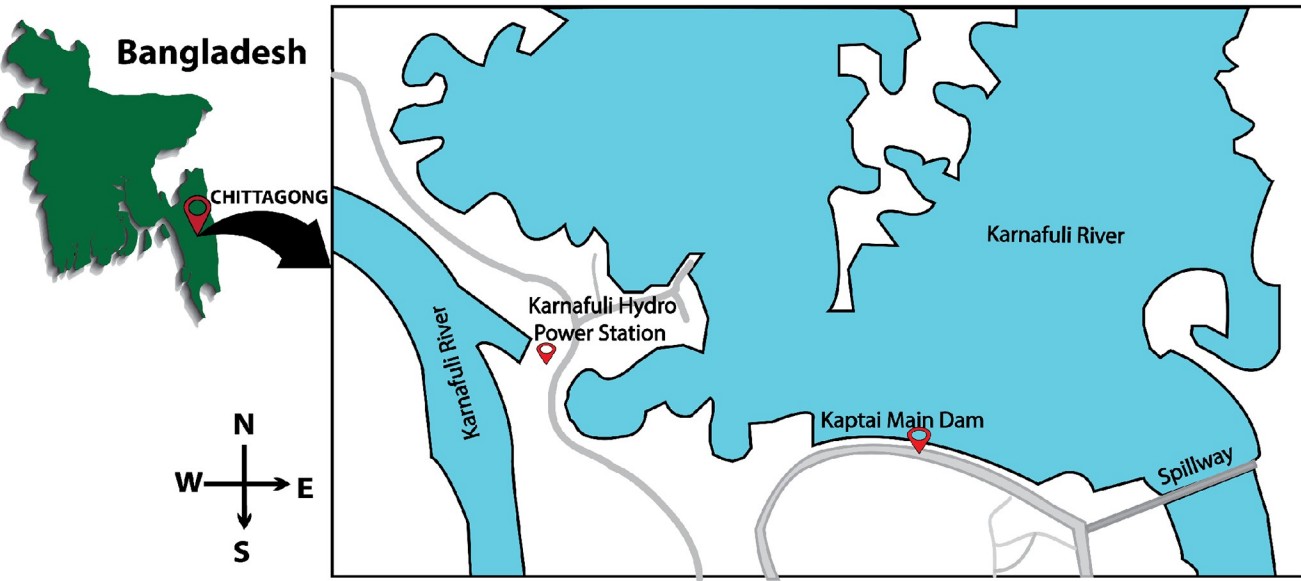

**Fig 1. The site map of Karnafuli Hydro Power Station (picture is drawn in Illustrator according to the geographical location).**

### 3.1 Location and capacity

Karnafuli Hydropower Station is the only hydropower plant in Bangladesh situated in the district of Rangamati and Chittagong. This power plant was assembled in the year 1962. The initial production was 80MW. After a couple of years, a 230MW capacity unit was installed to increase the capacity. There are four units installed currently in the KH plant, and most of the time in a year, all four units are not functioning due to lack of water flow. As the only hydro-based electricity generation in BD, KH plays a vital role to meet the demand of electricity of this region and to maintain the ecological balance. The massive reservoir of Kaptai provides flood management facilities downstream for the south side of the country. Fig 1 displays the map of the plant. The map shows the Karnaphuli river and the Kaptai main dam with the spillway. The hydro turbine of the KH plant is in the west side of the main dam.

### 3.2 Installed equipment specifications

Specifications of the installed equipment are provided in Table 1. All components are from Hitachi and Toshiba, Japan. Tables present the type and the rating of each installed component with the date.

## 4 Statistical data collection and analysis

### 4.1 In field data collection

The survey in the KH plant was conducted in the year 2020. Data were collected from unit-4 of the plant. It focuses on turbine discharge rate per day to know the feasibility of installing an extra turbine in the tailwaters. Fig 2 shows the volume of turbine discharge rate per second in a day in the peakiest month of July. The discharge rate is almost flat except for a bit of fluctuation. The overall output on the statistics of energy generation and turbine discharge data of unit-4 in the twelve months of the year 2020 is presented in Table 2. The power output of the

**Table 1. Specification of installed turbine, transformer, exciter, governor in KH plant.**

| Hydraulic turbine (Hitachi Ltd., Japan) | | Main transformer (Hitachi Ltd., Japan) | | Exciter (Toshiba, Japan) | |
|---|---|---|---|---|---|
| Type | Kaplan turbine | Type | Outdoor oil air cooled 3 phase | Type | Silicon controlled rectifier |
| No. of blades | 5 | Power | 62500 kVA | Rated excited voltage | 195 V DC |
| Speed | 136.4 rpm | Secondary | 132 KV Y | Rated rectified current | 645 A DC |
| Mfg. No | 144971-1 | Primary | 11 KV Δ | Celling voltage | 330 V DC |
| Date | 1986 | Governor, E-G-Type electro hydrolic | | Pressure | 50 $Kg/cm^2$ |
| **Output power in terms of Head** (Hitachi, Japan) | | | | | |
| | | Net head | | Output | |
| Maximum | | 30.94 m | | 52000 kW | |
| Rated | | 25.91 m | | 52000 kW | |
| Minimum | | 17.98 m | | 26000 kW | |

plant can be estimated by (1).

$$P = \frac{1}{2}\eta\rho A V^3 \tag{1}$$

## 4.2 Analysis

Table 2 shows the average energy and average turbine discharge rate during twelve months of operation in the KH plant in the year 2020. The average turbine discharge rate for twelve months is 76.36 m³/s, and the peak rate is 198.83 $m^3/s$ in July. Most importantly, the discharge rate shows almost a constant pattern throughout the month of July (Fig 2). Other months in the year have relatively more fluctuations. The minimum discharge rate is recorded in December, as this is the winter season in BD. There is a factor that exists between the energy generated and the turbine discharge rate. This factor follows almost similar trend over the year.

## 4.3 Existing model of Karnafuli Hydroelectric Power Station (unit-4)

The current layout of the existing model of Karnafuli Hydroelectric Power Station of unit-4 is displayed in Fig 3. The penstock draws water from the reservoir and injects it into the turbine blades, which results in turbine rotation. The turbine shaft is coupled with an AC Generator through a shaft. The scroll case with the turbine ensures uniform water flow towards the turbine blades. The turbine discharged water follows (as can be seen as region-1 in Fig 3) into the draft tube and finally exits to the tailrace. The governor system regulates the water stream flow to the turbine. Based on the in-field data analysis, a MATLAB simulation is performed to identify the estimated output of the existing plant of unit-4. Design specifications of the KH plant are displayed in Fig 4 as nomenclature. We used a Hydraulic Turbine Governor (HTG) block, a servomotor, and a proportional integral derivative (PID) governor system in the MATLAB Simulink to simulate the unit-4, as can be seen in Fig 5. HTG block is used to implement a nonlinear hydraulic turbine model. We set the reference speed to 1 pu and reference mechanical power to .8 pu. Intending to implement a DC exciter, we used an excitation system block in which the initial terminal voltage and field voltage are set to 1 and 1.3 pu, respectively. We used a synchronous machine block and operated it in generator mode, which takes mechanical power as input from the HTG system. This three-phase generator rated 62500 KVA, 11 kV, is connected to a 132 kV network through a Δ − Y 62500 KVA transformer. The system starts in a steady state, with the generator supplying 50 MW of active power. A PLL system is used to

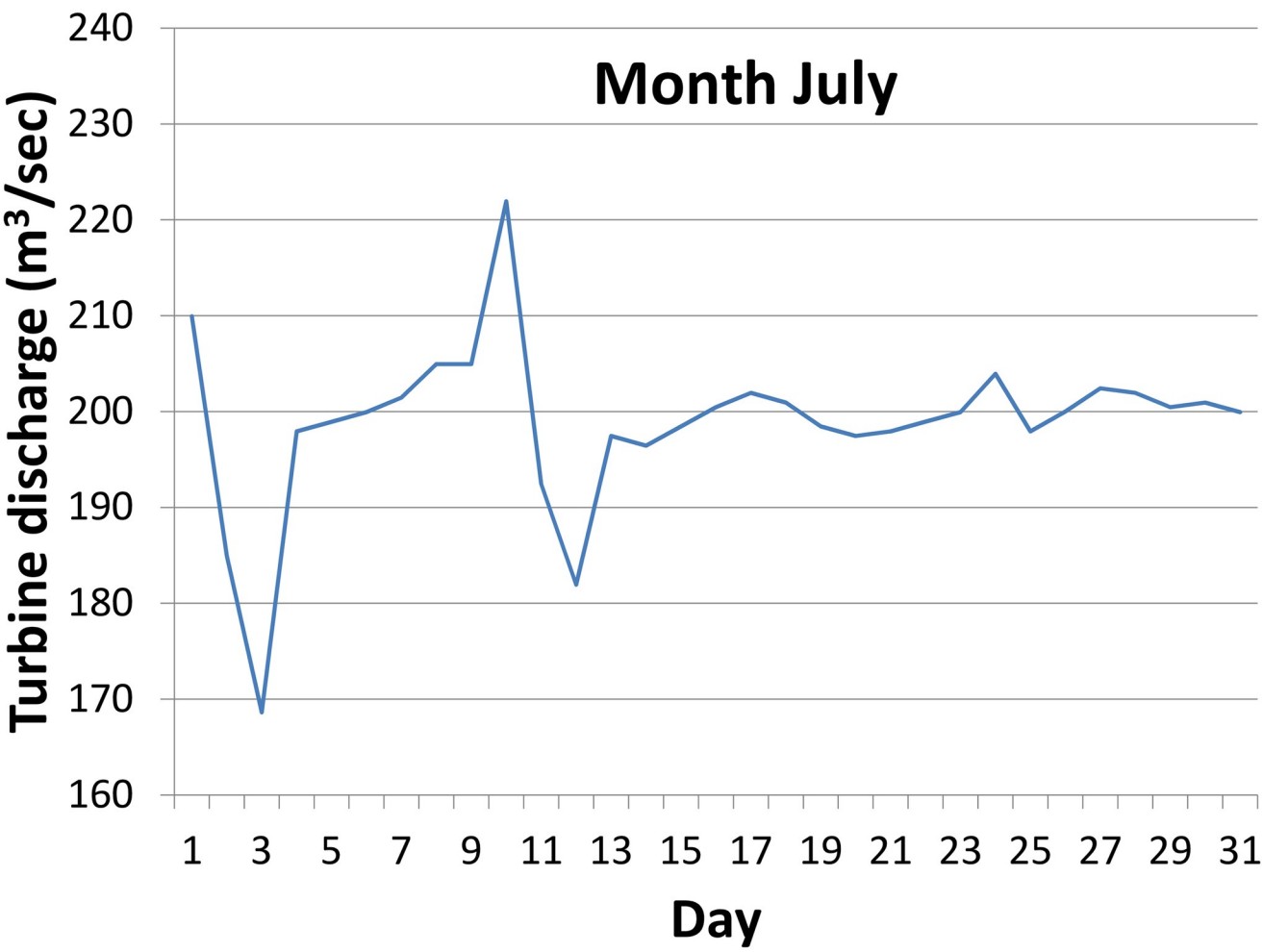

**Fig 2. Volume of turbine discharge rate in the month of July, 2020.**

**Table 2. Statistics of the average energy generation and turbine discharge rate.**

| Month | Average Energy Generated (J) | Average turbine discharge (cusec) | Average Turbine Discharge ($m^3/sec$) |
|---|---|---|---|
| January | 109435.48 | 649.35 | 18.39 |
| February | 145185.21 | 1102.51 | 33.72 |
| March | 266716.13 | 1945.32 | 55.09 |
| April | 840316.67 | 5476.93 | 155.11 |
| May | 360125.81 | 2619.26 | 74.18 |
| June | 509368.33 | 3196.63 | 90.53 |
| July | 1174603.23 | 7020.97 | 198.83 |
| August | 708074.19 | 4178.48 | 118.33 |
| September | 257690.45 | 1874.26 | 53.08 |
| October | 373216.13 | 2160.45 | 61.18 |
| November | 350226.67 | 2010.07 | 56.93 |
| December | 3229.03 | 36.16 | 1.02 |

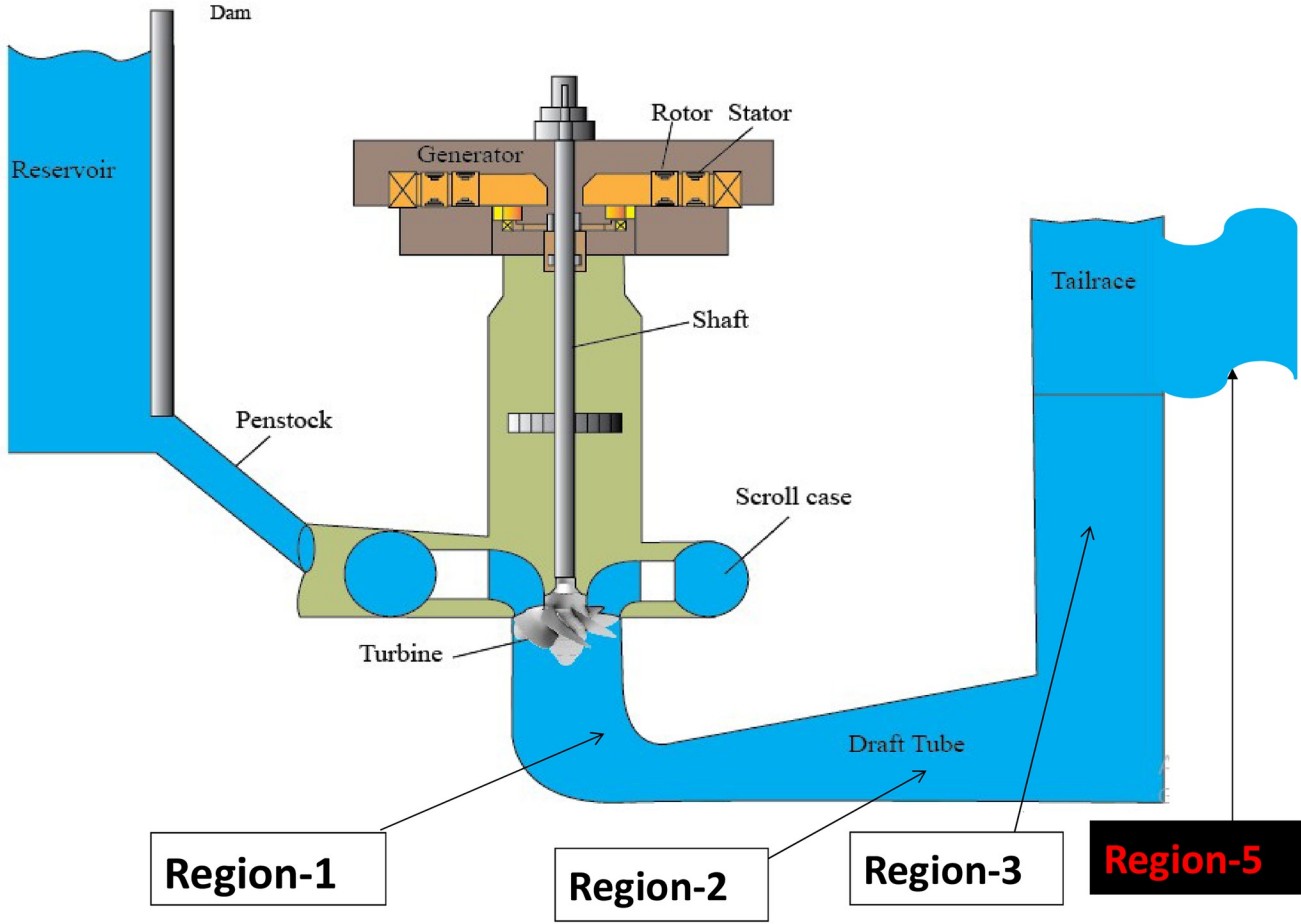

**Fig 3. Unit-4 model of Karnafuli Hydroelectric Power Station.**

synchronize on a set of variable frequencies, three-phase sinusoidal signals. If the Automatic Gain Control is enabled, the input (phase error) of the PLL regulator is scaled according to the input signal magnitude.

## 4.4 Accuracy of simulation model

The results of the simulated model of the KH plant are analyzed to design the proposed hydro-kinetic turbine in region-4 of Fig 3. Four different graphs of excitation voltage, the speed characteristics, the stator current, and the voltage output characteristics are displayed in Fig 6. A three-phase short circuit fault is introduced into the model to determine the system's response during and after the fault conditions. An analysis of the effectiveness of the entire network is conducted to justify the system's stability under faulty conditions. The simulation time for all simulated outputs is set to 2*sec*. Fig 6(a)–6(f) represents the waveform of different entities with stability.

A three-phase to ground fault is inserted into the model at a time of 0.1*sec*. The system experienced a steady-state condition at the initial stage of the simulation with an excitation voltage of 1.2pu (per unit) and an output voltage of 0.8pu. Under the faulty condition, the nominal speed is 1pu, and the stator current is 0.8pu. The inserted three-phase fault is continued for another 0.1*sec* and the system is back to a stable state at 0.2*sec*. The system experiences

| Hydraulic Turbine and Governor | Gate opening limit in pu | $g_{min}$: 0.01; $g_{max}$: 0.97518 $v_{gmin}$: -0.1; $v_{gmax}$: 0.1 |
|---|---|---|
| | Initial mechanical Power | 0.801828 pu |
| Excitation System | Low pass filter time constant | $20 \times 10^{-3}$ S |
| | Initial values of terminal voltage and field voltage | $V_{t0}$ = 1 pu $V_{f0}$ = 1.33337 pu |
| Synchronous machine | Nominal power, line-to-line voltage, frequency | Pn: $62.5 \times 10^{6}$ VA Vn(rms): 11000 V fn : 50 Hz |
| | Active Power Generation | P = 50 MW |
| Transformer | Primary (Δ) | $V_1(\emptyset - \emptyset, rms)$= $11 \times 10^{-3}$ V $R_1$: 0.0027pu $L_1$: 0.08pu |
| | Secondary (Y) | $V_2(\emptyset - \emptyset, rms)$=$132 \times 10^{3}$ V $R_2$ : 0.0027 pu $L_2$ : 0.08 pu |

**Fig 4. Nomenclature of design specifications for simulating the existing model of unit-4 of the KH plant.**

a significant drop in the output voltage of 0.3pu, and the generated voltage $V_a$ becomes stable after 0.2*sec*. In addition, there is an increment in generator stator current to 4.2pu under a transient state at 0.2*sec*, and it becomes stable at 0.4*sec*. Furthermore, the excitation voltage of the system increases drastically to a value of 11.7pu, and the result of the rate of valve opening and closing of the governor speed rose to 1.01pu in the same direction.

It is well known that the speed of a generator varies with the change of its flux density. A similar case happens when the inserted fault causes an increased flux density in the generator field. Due to this increased flux density, the speed and the terminal voltage of the generator are increased significantly. Subsequently, a rise in flux will have the effect of bringing the terminal voltage back to its initial value because the fault immensely reduces it. The oscillation of the speed does not return to its initial state of the hydraulic turbine. Under these circumstances, the speed of the rotor becomes unstable during and after the fault initiation. The excitation voltage returns to stable state after a period of 0.5*sec*. The generator's terminal voltage is stepped up to 100KV with the current value of 190 A by a step-up transformer. The operating frequency of 50*Hz* is obtained. The analysis depicts that the model can tolerate short circuit faults and can be back to stable condition in time.

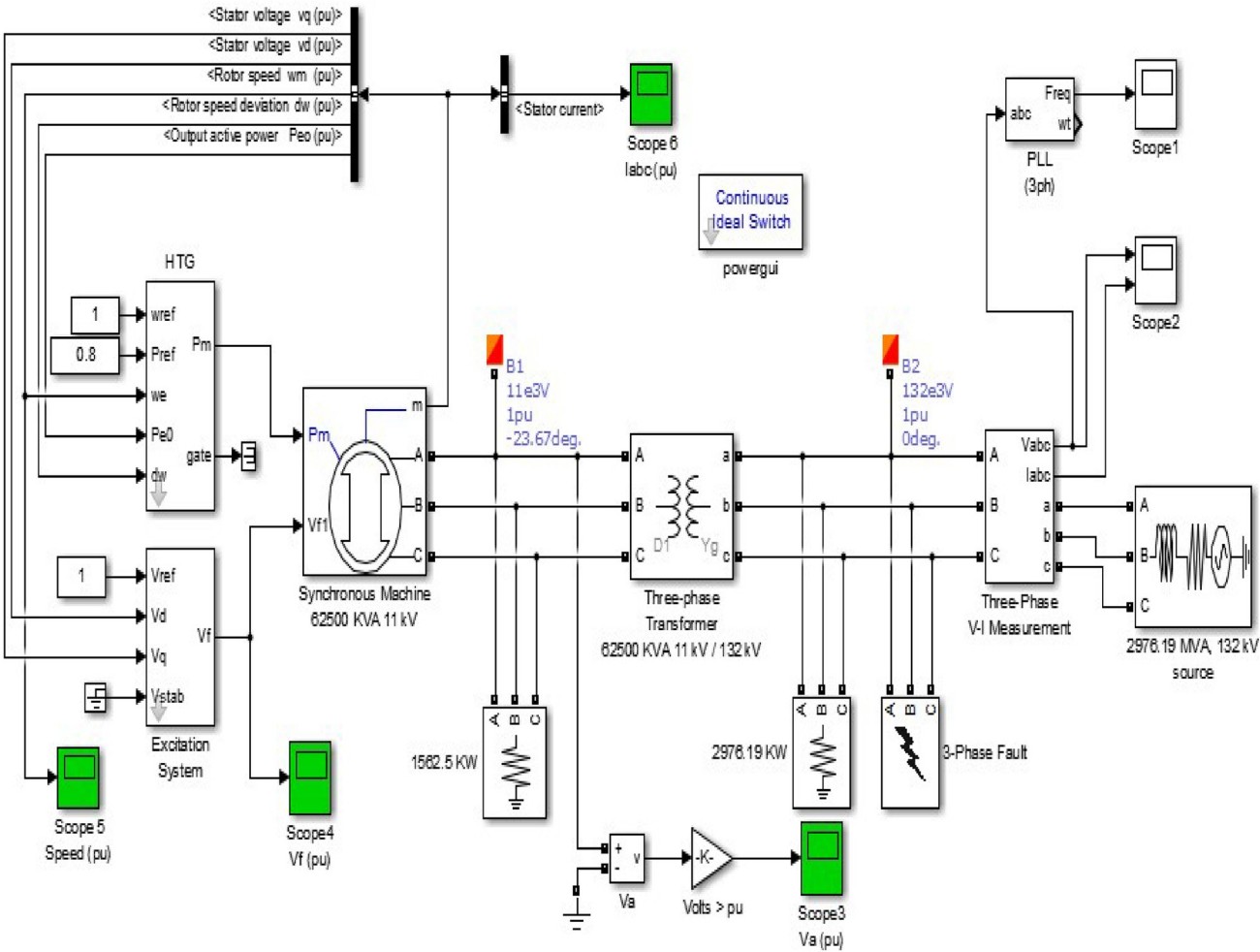

**Fig 5. Simulation of Karnafuli Hydroelectric Power Station (unit-4).**

## 5 Proposed hydrokinetic method

We propose a hydrokinetic turbine in series with the hydrostatic turbine under the combined cycle hydropower methodology to increase the power output of the KH plant. The possible means of increasing the efficiency of the existing power station is to model and simulate the generating stations, which aids in describing the benefits of integrating another technology making a combined cycle power plant. This part describes the dynamic model of the proposed technique to extract hydrokinetic energy from the tailwater of the station.

### 5.1 Identification of extraction region

A feasibility study is conducted to propose an efficient design of the hydrokinetic turbine based on data analysis. The focused area is to identify the optimum region to install the turbine. The possible area of interest is runner output after the powerhouse in the tailwater. The possible site for obtaining the maximum water velocity and absolute pressure is crucial for designing the kinetic turbine.

In Fig 3, it divides the entire plant into four regions of interest to find out the best possible zone to install the hydrokinetic turbine. Region-1 is the semi-circular area of the runner after

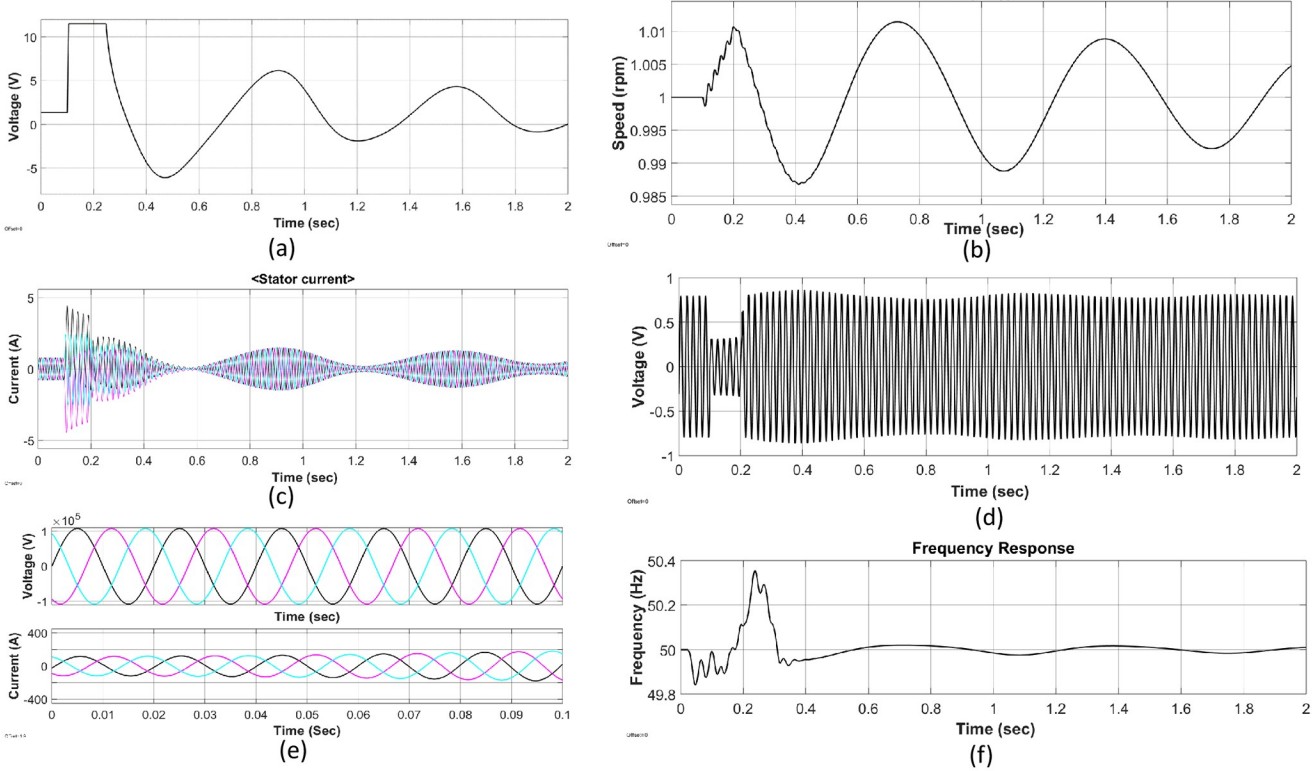

**Fig 6. Outputs of the simulation results of unit-4: (a) waveform of excitation voltage, (b) speed of the rotor, (c) stator current, (d) generated voltage, (e) 3-phase voltage and current at load side and (f) overall frequency response.**

the hydrostatic turbine discharge path. In this region, water from the main turbine discharges rapidly, and the energy remains static, thus no possibility of considering the kinetic energy source. Midway, thru elbow region, is identified as region-2 in the draft tube. It is possible to place a hydrokinetic turbine in region-2 as a colossal water current is maintained. However, the draft tube functions like a turbine discharge outlet by which an expansion principle is maintained to keep an efficient water discharge rate. If a barrier is placed in region-2, the primary turbine efficiency is reduced. In addition, the KH plant water head is measured to the draft tube exit point. This may affect the actual output of the plant. One possible way to consider requires a massive engineering reconstruction in this region which is not cost-effective.

Therefore, attention can be given to the remaining two regions for consideration. The region-3 is downstream of the pier nose inside the bay outlet area to the tailrace. The direction of the water current is opposite to the direction of the region-1. For a hydrokinetic turbine, this area is not suitable; rather hydrostatic turbine is desirable. Massive reconstruction is required in this region for installing such turbines. The last region is the tailrace, in which a significant horizontal water current is produced in the KH plant. Therefore, region-4 is proposed as the hydrokinetic turbine installation site.

Data analysis from Table 2 shows that region-4 can harness the maximum augmented output power. It considers a certain distance from the draft tube exit point to place the proposed turbine by building an artificial channel in the tailrace, as can be seen in Fig 7. In the proposed design, this artificial channel contains an undershoot hydrokinetic turbine.

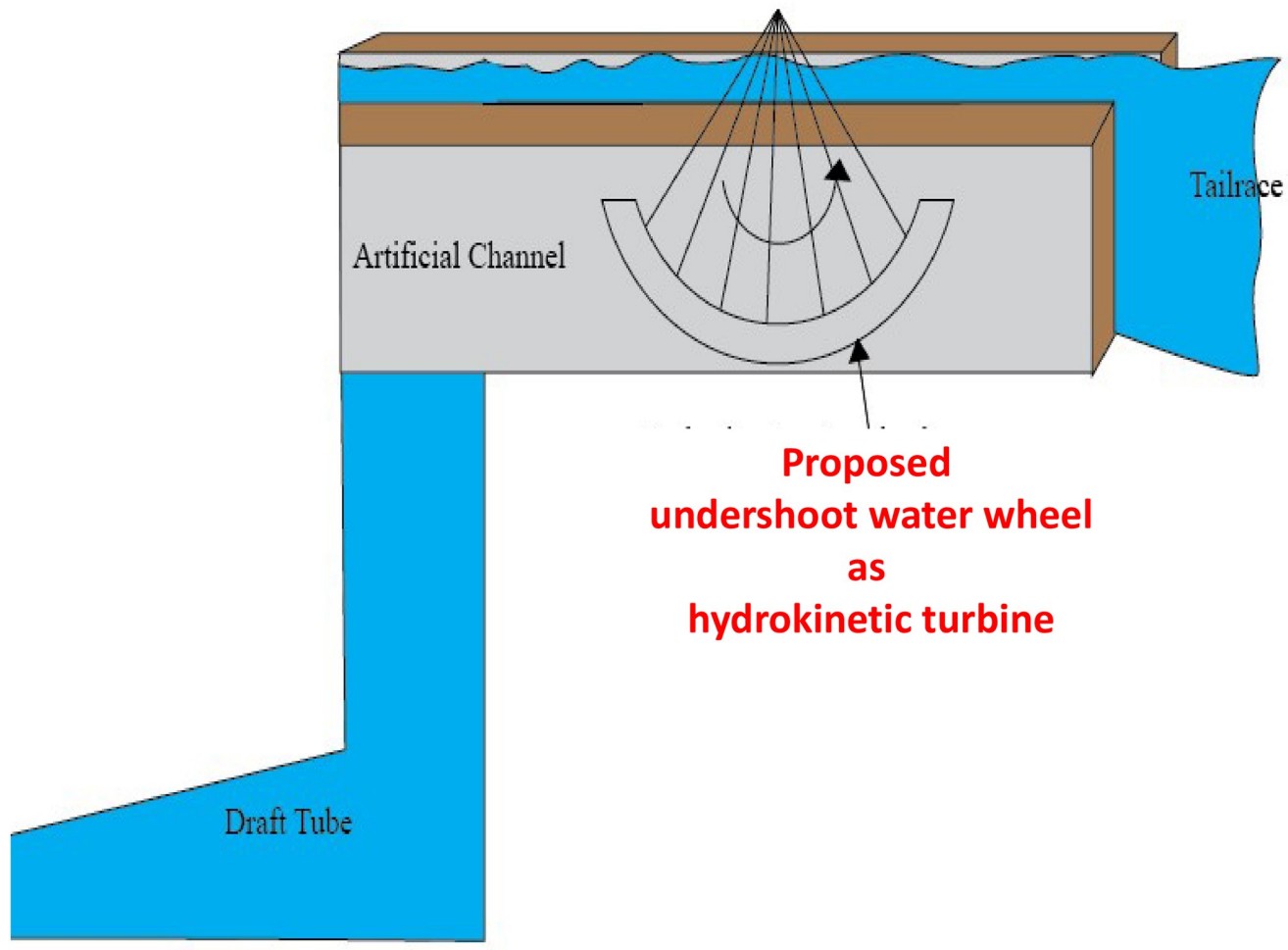

**Fig 7. Proposed artificial channel construction to install hydrokinetic water turbine.**

## 5.2 Prototype model

The proposed prototype model consists of a hydrokinetic turbine, alternator, and a power-house to accommodate the alternator with a gear coupler of the turbine. The complete model is shown in Fig 8.

**5.2.1 Design of proposed model: Augmented power generation.**   Based on the in-field survey of the KH plant from Table 2, it considers different parameters such as channel radius, length, inside-outside area of the channel, turbine blade number, gear characteristics, and generator specifications as shown in Fig 8.

The proposed model is designed in Matlab Simulink with the considered specifications. Fig 9 displays the proposed model. PMDC generator, battery, and inverter circuit to invert DC to AC output are used for the generating station in the simulation. It considers the nominal phase to phase voltage as 220V with a 50Hz frequency. The three-phase series inductor and capacitance are 2.97 mH and 5.33 mF, respectively, with armature resistance of 0.6 ohms. The generator indicated on the right side of Fig 9 takes rotor speed as the input given by a constant value of 7.89, calculated from the average turbine discharge water flow found at the tailrace area. An inverter unit converts the generated DC to AC power. A universal bridge block with a

| Symbol | Name | Value |
|--------|------|-------|
| $A_{Ch}$ | Channel Area | 7 m² |
| $r_{Ch\_out}$ | Outside radius | 1 m |
| $r_{Ch\_in}$ | Inside radius | 0.5 m |
| $S_{Ch}$ | Cross sectional area | 0.523 m² |
| $\theta_{TB}$ | Angle of turbine blades | 1.047 rad |
| $r_{G\_big}$ | Big gear radius | 0.495 m |
| $r_{G\_small}$ | Small gear radius | 0.22 m |
| $T_{G\_big,}$ $T_{G\_small}$ | Teeth numbers of the gears | 198, 22 |

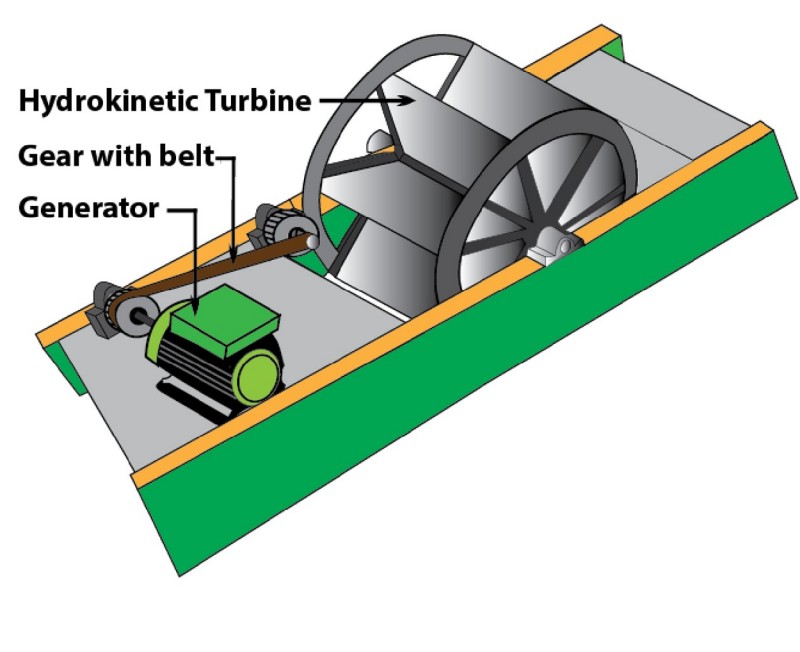

**Fig 8. Proposed extension unit in the KH plant with parameters.**

universal three-phase power converter that consists of up to six power switches connected in a bridge configuration is used. We used a PWM Generator to control the converter, which connects the universal bridge block to it. The Battery block implements a generic dynamic model parameterized to represent the most popular types of rechargeable batteries. The system is designed in such a way that it can supply the local baseload. In the simulation, we put three different scopes to observe other parameters.

**5.2.2 Mathematical model.** In the proposed turbine design, the water current direction in the artificial channel is considered horizontal, which means there is no gravitational torque. In Fig 7, the placement of the hydrokinetic turbine is shown. To obtain optimum efficiency of the design, turbine, generator, and other specified components are given in Fig 8. The water current velocity, turbine discharge rate, and power output of the generator can be obtained by the following equations based on the specifications.

Considering the tidal velocity $V_{tidal}$ and water velocity after turbine $V_{after\_t}$, the water flow mass in the artificial channel can be written as in (2). Assuming the velocity coefficient is 0.33, $V_{after\_t}$ can be obtained from the angular velocity (w) and radius ($R_{turbine}$) of the turbine by considering $V_{after\_t}$ is equal to $wR_{turbine}$, which implies $0.33V_{tidal}$.

$$m = \rho A_{ch}(V_{tidal} - V_{after_t}) \tag{2}$$

The water force ($F_W$) per unit of time $t$ is received by the turbine blades can be written as in (3).

$$F_W = \frac{d}{dt}[m(V_{tidal} - V_{after_t})] = 0.44\rho A_{ch} V_{tidal}^2 \tag{3}$$

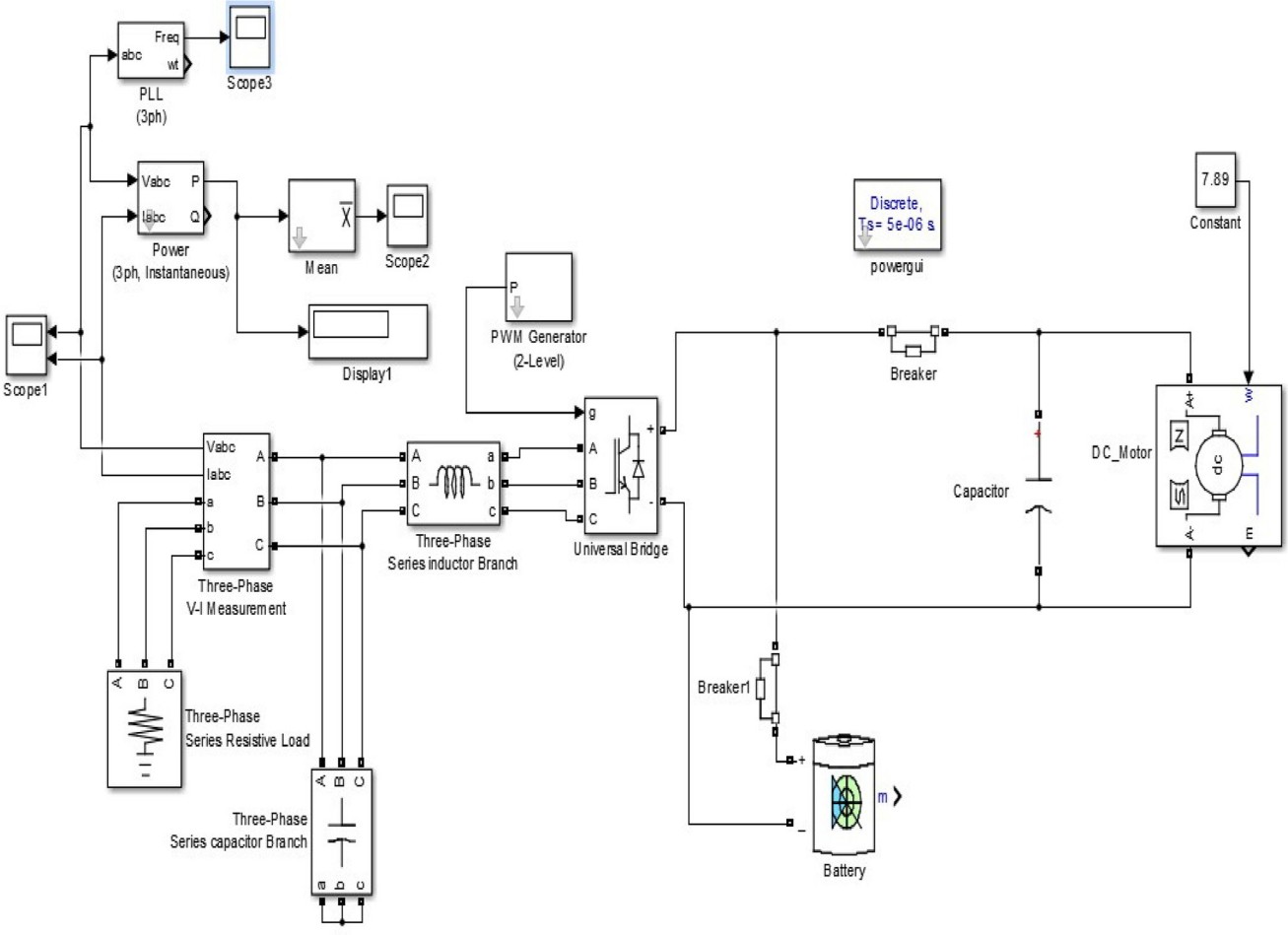

**Fig 9. Simulation based model to verify the design integrity.**

If the turbine output power is $P_{turbine}$ and can be obtained from $P_{turbine} = F_W * V_{after\_t}$, which is $0.145\rho A_{ch} V_{tidal}^3$ according to (3). Similarly, the input power $P_{in}$ into the turbine is $1/2 * A_{ch} * V_{tidal}^2 dx/dt = 1/2\rho A_{ch} * V_{tidal}^3$ as the rate of change of distance $x$ is the tidal velocity of water. Therefore, the efficiency $\eta$ of the turbine is estimated by the ratio of the output to the input of the turbine, which is 0.269 or approximately 27%. On the other hand, water current velocity $V_{Wcurrent}$ can be determined by the ratio of turbine discharge quantity $Q_{turbine}$ to the exit area $A_{ch}$ of the draft tube. According to Betz's law, output power and turbine velocity can be written as (4) and (5) respectively.

$$P_{out} = \frac{1}{2}\rho A_{ch} V_{turbine}\left(V_{tidal}^2 - V_{after_t}^2\right)$$
$$= \rho A_{ch} V_{tidal}^2\left(V_{tidal} - V_{after_t}\right)$$
(4)

$$V_{turbine} = \frac{1}{2}\left(V_{tidal} + V_{after_t}\right)$$
(5)

# 6 Results

## 6.1 Estimation from surveyed data

In a hydrokinetic power generation system, especially from the draft tube outlet of the hydropower plant, water flows in the horizontal direction. In this case, no gravitational torque will work. Therefore, to extract this energy properly, it is needed to build a system in such a way by adjusting the water turbine and an alternator system. An alternator system is essential for converting mechanical energy into electrical energy. Perfect adjustment of the turbine blade is necessary for optimum efficiency. Utilizing the 12 months data from Table 2 and putting them into (1), the hydrokinetic energy is estimated. The total output power is estimated from 98 kV and 151A of a power factor of 0.9. The calculated results are demonstrated in Table 3. This result displays the capability of generating an average of 13.32 MW of energy by the proposed method.

## 6.2 Simulation result of proposed model

The average output power of the proposed simulation model in Fig 8 is 12.2 MW which is much closer to the theoretically calculated value (13.32 MW) in Table 3. The generating station comprises of PMDC generator, battery, and inverter circuit. The power generated by the generator is DC. Because most of our home appliances are AC loads, it needs to be converted to AC; thus, an inverter circuit is used. The load terminal voltage is obtained 7kV under different loading conditions such as resistive, inductive as well as capacitive, as can be seen in Fig 10. The load current is obtained as 1500A with a frequency of 50 Hz. The system is designed in such a way that it can supply the local baseload. During the off-peak period, it stores energy through the battery. The batteries deliver at the time of the peak load period to continue the service.

## 6.3 Experimental result

The experimental setup is demonstrated in Fig 11. A pilot prototype on a tiny scale is presented to verify the functionality and estimation of the output power of our proposed technique. All equipment is managed from the local market at a significantly lower price. Some of them are

**Table 3. Estimation of output power using historical data of water discharge.**

| Month | Water Velocity (m/sec) | Hydrokineticwater current power (watt) |
|---|---|---|
| January | 2.63 | 63670.06 |
| February | 5.32 | 1253031.75 |
| March | 7.87 | 1706051.91 |
| April | 22.16 | 38087047.94 |
| May | 10.6 | 4168556 |
| June | 12.93 | 7565952.65 |
| July | 28.4 | 80172064 |
| August | 16.9 | 16893831.5 |
| September | 10.69 | 5750345.03 |
| October | 8.74 | 2336696.68 |
| November | 8.13 | 1880787.29 |
| December | 0.15 | 11.81 |
| Average power (estimated) | | 13323170.55 (13.32 MW) |
| **Average power (Simulated)** | | **12203487.13 (12.20 MW)** |

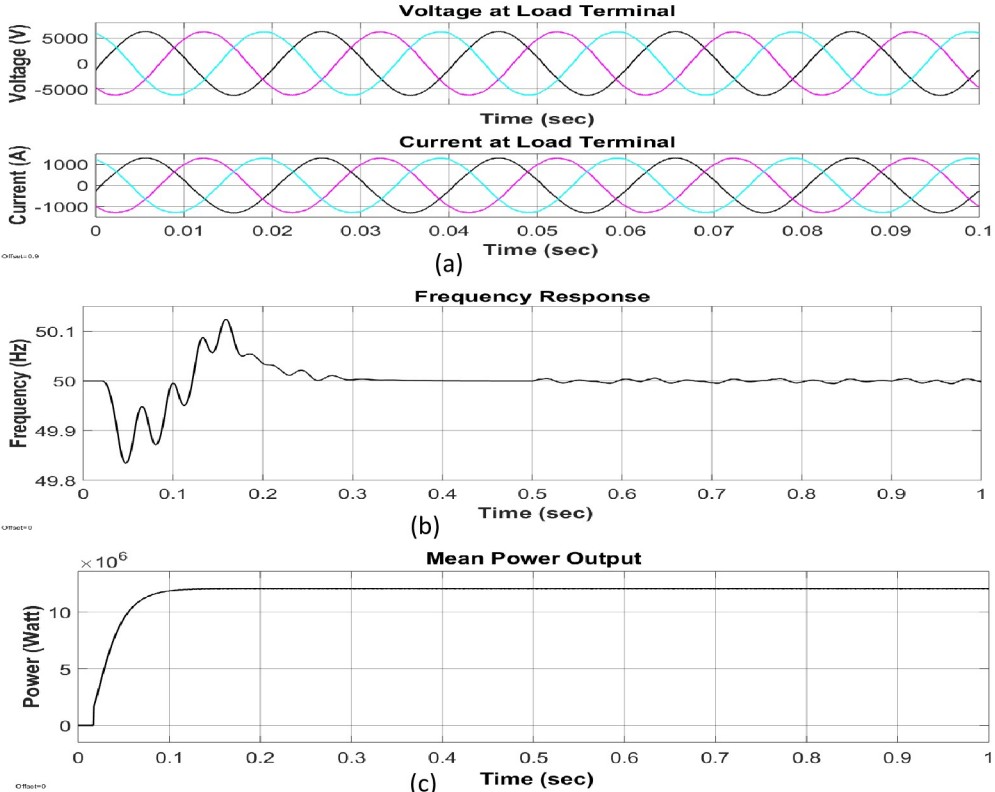

**Fig 10. Output of our proposed hydroturbine: (a)three-phase parameter of voltage and current, (b)frequency response of the power output, (c) Waveform of mean output power.**

unused parts of other devices. In this experiment, we used a bucket of 30 liter, which is analogous to the reservoir of a hydropower station. Though the reservoir's capacity is too small compared to a substantial hydropower water storage, we tried to prove the practicability of our proposed hydrokinetic power model by implementing a prototype model on a small scale by using the equipment which is available in the nearest marketplace. A DC fan is used as a hydrostatic turbine analogous to the Kaplan turbine of the KH plant. We placed the reservoir at the height of 48 inches above the hydrostatic turbine, which means the water head of the model is 48 inches. A polyvinyl chloride (PVC) pipe is used to pass the water from the reservoir towards the hydrostatic turbine, which performs the function of the penstock. To control the release of water as well as increase the pressure of water, we used a water tap through which water is injected into the turbine. The turbine discharge water enters the draft tube. A tailrace is required to discharge the water leaving the turbine into the river. The draft tube must remain water-sealed all the time. The penstock and draft tube diameters are $25.4cm$ (1 inch) each. The hydrokinetic turbine inlet and outlet areas are $3.98cm^2$ each.

In this prototype model, we use a undershoot water turbine as a hydrokinetic turbine. This type of turbine is appropriate for a particular area (in our case, it is region-4) because there exists a horizontal flow of water. The amount of water that is injected into the first turbine, the same quantity of water leaves from the draft tube as it is an enclosed linear system. There is a significant water speed at the draft tube exit point that is the kinetic energy. This energy is extracted from the second turbine installed at the draft tube exit point.

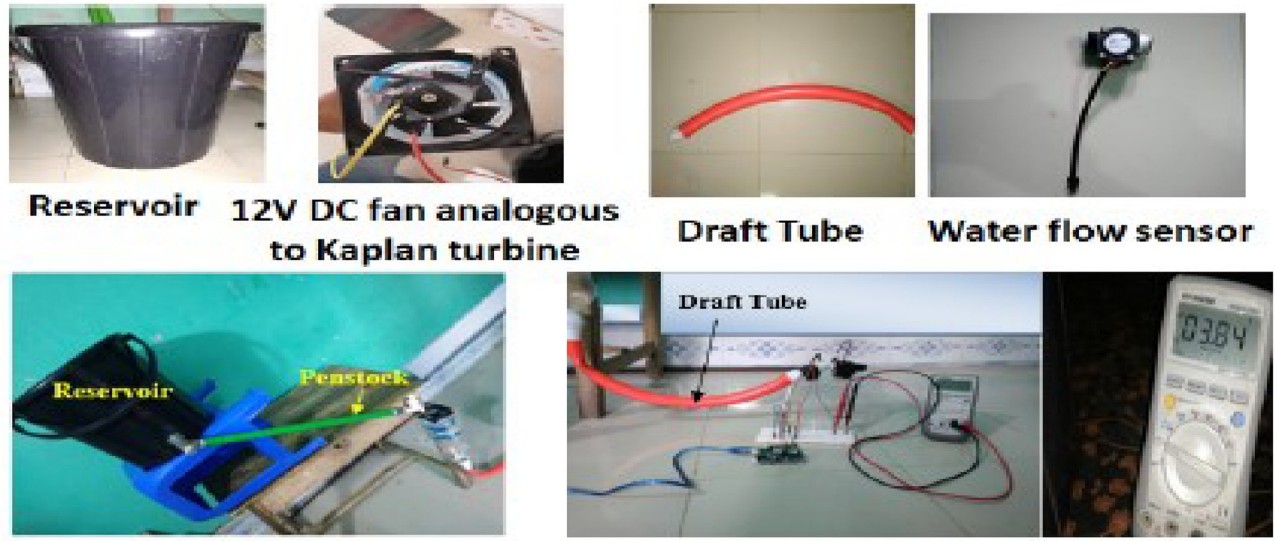

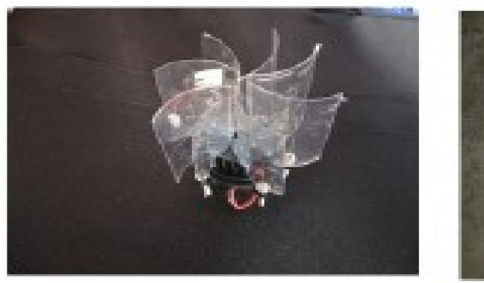

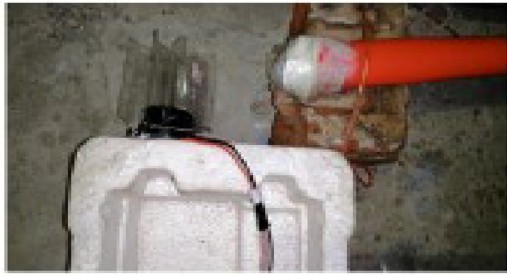

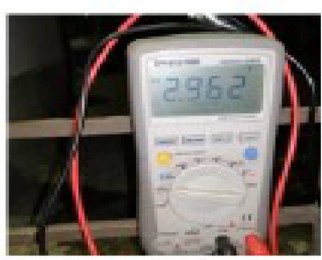

**Fig 11. Proposed prototype model is in a small scale.**

The average output from the hydrostatic turbine is 3.8V, and it is 2.9V from the hydrokinetic turbine. The average power generated from the proposed hydrokinetic turbine is 0.78 watts with 324 mA current and 0.9 power factor. As the experimental setup is on a small scale in our prototype, we can extend it on a larger scale, for example, 1000 times. In this case, the projected average output from the hydrostatic turbine will be (3.8 V X 1000) 3800V with an average output of (2.9 V X 1000) 2900V from the hydrokinetic turbine. The output of the prototype provides stable output without any distortion with a frequency of 50 Hz.

## 6.4 Discussion and comparison

This work focuses on the potential and feasibility of introducing a hydrokinetic energy generation system in the KH plant. Annually, an average of 13.32 MW water current power is estimated from the practical surveyed data. Table 4 displays a comparison among the estimated, simulated, and experimental results. The estimated and simulated power are very closed to each other. The prototype model output power is reported 0.78 Watt with proper output voltage and stability responses. This small-scale power can be boosted to the simulated output

**Table 4. A comparison among estimated, simulated and experimental results.**

| Parameters | Simulated (Unit-4 of KH) | Estimated (from Data) | Simulated (proposed) | Prototype Model |
|---|---|---|---|---|
| Voltage | 100kV | 98 kV | 7 kV | 2.9 V |
| Current | 190A | 151 A | 1500A | 324 mA |
| Frequency | 50 Hz | 50 Hz | 50 Hz | 50 Hz |
| Average, P | 17.1 MW | 13.32 MW | 12.20 MW | 0.78 Watt |

(12.2 MW) level if it can be implemented on a bigger scale. Our investigation result showed enough possibilities to generate hydrokinetic energy at the draft tube exit point and tailrace area in the hydropower plant. Placing an additional turbine at that selected region can create extra 12.2MW power around 27% of existing plant capacity. The proposed approach used an undershoot water turbine for this purpose as the water flows along the horizontal axis at that region. Our prototype model and simulation results demonstrate the practicability of the method. Our results show that it is possible to generate large-scale additional power at a low cost by adding the hydrokinetic turbine to the existing conventional hydropower.

## 7 Conclusion and recommendations

In this work, a survey has been performed in an existing power plant in Bangladesh and a method has been proposed to increase its power output to attain sustainability in the nearby regions. The result shows that 12.2 MW additional power can be generated from the KH plant with a minimal cost. It can be stated that the contribution in hydro energy can open a new window for achieving SDGs in Bangladesh especially, SDG7. The CCHS method utilizes the exit stream of the traditional hydropower plant but do not change the characteristic pathway of the stream in a significant way. Subsequently, it can supply additional clean energy with fewer prerequisites for construction work, less influence on the environment, minimal noise and aesthetics issues. The outflow of the dam varies a little, which is predictable and therefore unlike the wind energy the output power can be estimated beforehand. Furthermore, the hydrokinetic turbine system would require less rigorous fast-acting control and protection techniques. The immediate implementation of this method by optimal turbine placement downstream of the hydropower plant, to extract the maximum amount of energy at least cost, can lead us one step forward for achieving SDG7. The technology required for extracting energy from draft tube exit water is still at a very early stage of development and more technical and feasibility investigation is required in this field. A more extensive feasibility analysis of CCHS would be required to develop a scaled-up conceptual system-level design, performance, and economic study of the combined generation and conventional hydropower facilities. An extensive study of a selected hydropower stations needs to be conducted corresponding to the site selection, environmental effect, hydraulic impact, and environmental impact assessment for augmenting it to a CCHS.

## Acknowledgments

The authors gratefully acknowledge to the Karnaphuli Hydropower Plant authority for providing different data sets. The authors also acknowledge to Bangladesh Army International University of Science and Technology (BAIUST), Bangladesh to conduct this research using all kinds of facilities of Electrical and Electronic Engineering department.

## Author Contributions

**Conceptualization:** Fakir Sharif Hossain.

**Data curation:** Fakir Sharif Hossain, M. Altaf-Ul-Amin.

**Formal analysis:** Fakir Sharif Hossain, Tahmidur Rahman, Omar Bin Mannan, M. Altaf-Ul-Amin.

**Investigation:** Tahmidur Rahman, Abdullah Al Mamun.

**Methodology:** Fakir Sharif Hossain, Tahmidur Rahman, Abdullah Al Mamun, Omar Bin Mannan.

**Resources:** Fakir Sharif Hossain, M. Altaf-Ul-Amin.

**Software:** Tahmidur Rahman, Abdullah Al Mamun, Omar Bin Mannan.

**Supervision:** Fakir Sharif Hossain, M. Altaf-Ul-Amin.

**Validation:** Fakir Sharif Hossain.

**Writing – original draft:** Fakir Sharif Hossain.

**Writing – review & editing:** Fakir Sharif Hossain, M. Altaf-Ul-Amin.

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
