## [Decision Letter · Decision Letter 0]

4 Jul 2021

PONE-D-21-18420

An Approach to Increase the Power Output of Karnafuli Hydroelectric Power Station: A Step to Sustainable Development in Bangladesh's Energy Sector

PLOS ONE

Dear Dr. Hossain,

Thank you for submitting your manuscript to PLOS ONE. After careful consideration, we feel that it has merit but does not fully meet PLOS ONE’s publication criteria as it currently stands. Therefore, we invite you to submit a revised version of the manuscript that addresses the points raised during the review process.

We look forward to receiving your revised manuscript.

Kind regards,

Mehrdad Ahmadi Kamarposhti

Academic Editor

PLOS ONE

Journal Requirements:

3. Please amend the manuscript submission data (via Edit Submission) to include authors Tahmidur Rahman, Abdullah Al Mamun, Omar Bin Mannan.

4. We note that  Figure1 in your submission contain map/satellite images which may be copyrighted. All PLOS content is published under the Creative Commons Attribution License (CC BY 4.0), which means that the manuscript, images, and Supporting Information files will be freely available online, and any third party is permitted to access, download, copy, distribute, and use these materials in any way, even commercially, with proper attribution. For these reasons, we cannot publish previously copyrighted maps or satellite images created using proprietary data, such as Google software (Google Maps, Street View, and Earth). For more information, see our copyright guidelines: http://journals.plos.org/plosone/s/licenses-and-copyright.

4.1.    You may seek permission from the original copyright holder of Figure 1 to publish the content specifically under the CC BY 4.0 license. 

4.2.    If you are unable to obtain permission from the original copyright holder to publish these figures under the CC BY 4.0 license or if the copyright holder’s requirements are incompatible with the CC BY 4.0 license, please either i) remove the figure or ii) supply a replacement figure that complies with the CC BY 4.0 license. Please check copyright information on all replacement figures and update the figure caption with source information. If applicable, please specify in the figure caption text when a figure is similar but not identical to the original image and is therefore for illustrative purposes only.

5. Please include a copy of Table 6 which you refer to in your text on page 10.

6. Please upload a copy of Supporting Information Figure/Table/etc. xxxx which you refer to in your text on page xx.

Reviewers' comments:

Reviewer's Responses to Questions

**Comments to the Author**

1. Is the manuscript technically sound, and do the data support the conclusions?

Reviewer #1: Partly

Reviewer #2: Partly

2. Has the statistical analysis been performed appropriately and rigorously? 

Reviewer #1: N/A

Reviewer #2: No

3. Have the authors made all data underlying the findings in their manuscript fully available?

Reviewer #1: No

Reviewer #2: No

4. Is the manuscript presented in an intelligible fashion and written in standard English?

Reviewer #1: No

Reviewer #2: Yes

5. Review Comments to the Author

Reviewer #1: The first and general comment relates to the written form, as the paper requires a serious improvement in English grammar and spelling. The English should be improved. A deep proofreading is needed, many errors can be seen through the document.

The introduction and the related review of the literature is poor provided. And the structure must be improved for a better understanding of the current state of the art. Also, the drawbacks of the existing methods must be highlighted clearly for justifying the upgrade proposed by the current work. The way the latter work is improving the state of the art must be clarified.

Novelty of the paper is not mentioned obviously. There're a grammatical and syntax errors.

Authors are encouraged to introduce a nomenclature section at the beginning of the manuscript, including all variables, acronyms, indexes and constants defined in the manuscript, in order to make the text more clear and readable.

In the introduction section, the authors need more to introduce the previous literature. A comprehensive paper needs more than 40 references at least.

Figures should be given with better accuracy and described in the paper. Figures must be replaced with high resolution ones.

Authors must be talk about the future work and potential limitations briefly in the Conclusions and Recommendations section.

Variables in the text must be italic.

All those comments are unfortunate when we see the quality of the numerical results. Via those results, the proposed method demonstrates its effectiveness without any doubt. However, the materials for introducing the state of the art and the methodology is deficient and poor.

Reviewer #2: An Approach to Increase the Power Output of Karnafuli Hydroelectric Power Station: A Step to Sustainable Development in Bangladesh’s Energy Sector

Although it was felt that your manuscript was quite interesting. The main problems with the paper are as follow:

1-First paragraph of the introduction needs to be rewritten. (You can omit this paragraph)

2-“In this work, a method is proposed to obtain additional power from the existing plant 41 utilizing the Combined-Cycle Hydropower System (CCHS) technology to increase overall plant capacity [12].” is it done in reference [12]?!

3-Introduction needs to be rewritten. Please review more papers and mention their method weaknesses.

4-"An Approach to Increase the Power Output of Karnafuli Hydroelectric Power Station" How much did it increase?

5-The main concern with this paper is the contribution of the paper. The mentioned contributions cannot be new idea.

6-Studies have been performed on the Karnafuli Hydroelectric Power Station. Please compare simulation results with practical results (if possible).

7-References should be written as standard

8-In addition, proofread the text to avoid fragments, compliance with English grammar and punctuations rules and subject and verb agreement.

6. PLOS authors have the option to publish the peer review history of their article (what does this mean?). If published, this will include your full peer review and any attached files.

Reviewer #1: No

Reviewer #2: No

---

## [Author Response · Author response to Decision Letter 0]

29 Jul 2021

Response to Reviewers

Editor Comments: 

 Addressed: 

Thank you. We have followed the style according to the style format of PLOS ONE. We used the Template for PLoS-Version 3.5 March 2018 Latex version. 

 I already have an ORCID ID.

3. Please amend the manuscript submission data (via Edit Submission) to include authors Tahmidur Rahman, Abdullah Al Mamun, Omar Bin Mannan.

 Addressed: We have included the co-authors.

4. We note that Figure1 in your submission contain map/satellite images which may be copyrighted. All PLOS content is published under the Creative Commons Attribution License (CC BY 4.0), which means that the manuscript, images, and Supporting Information files will be freely available online, and any third party is permitted to access, download, copy, distribute, and use these materials in any way, even commercially, with proper attribution. For these reasons, we cannot publish previously copyrighted maps or satellite images created using proprietary data, such as Google software (Google Maps, Street View, and Earth). 

Addressed: 

We are extremely sorry for the copy right issue. We have tried to take the Google permission but it seemed to taking time. We redraw the Figure-1 in Illustrator to display the site of the KH plant.

5. Please include a copy of Table 6 which you refer to in your text on page 10. 

 Addressed: 

Thank you for the comment. It was a typo, sorry. It will be Table-3. We have corrected it.

6. Please upload a copy of Supporting Information Figure/Table/etc. xxxx which you refer to in your text on page xx.

 Addressed:

The supporting information that presented in the manuscript were the same Figures and Tables in the manuscript. Therefore, no additional data or information remains. We have omitted this part from the manuscript.

Reviewers' comments:

Reviewer's Responses to Questions

Comments to the Author

1. Is the manuscript technically sound, and do the data support the conclusions?

Reviewer #1: Partly

Reviewer #2: Partly

Ans: Thank you for the comments.

2. Has the statistical analysis been performed appropriately and rigorously?

Reviewer #1: N/A

Reviewer #2: No

Ans: Thank you for comments. We have data collection phase and simulation results to show the consistency of the existing KH plant output compared to our estimation theoretically.

3. Have the authors made all data underlying the findings in their manuscript fully available?

Reviewer #1: No

Reviewer #2: No

Ans: Thank you for comments. Actually we have provided all data and information in the corresponding Tables and Figures. We mistakenly added some supporting information data, but those are same figures and Tables in the manuscript. Therefore, all data are within the manuscript. Sorry for the inconvenience.

4. Is the manuscript presented in an intelligible fashion and written in standard English?

Reviewer #1: No

Reviewer #2: Yes

Ans: Thank you for comments. We have rewritten the manuscript thoroughly and added more sections to present in more readable form. We added subsection 1.1 in page-2 under the title of related works and Motivation.

5. Review Comments to the Author

Reviewer #1: 

The first and general comment relates to the written form, as the paper requires a serious improvement in English grammar and spelling. The English should be improved. A deep proofreading is needed, many errors can be seen through the document.

Addressed: Thank you for your comment. We have checked the manuscript thoroughly and try to improve the writing. The proofread has performed in peer.

ii)The introduction and the related review of the literature is poor provided. And the structure must be improved for a better understanding of the current state of the art. Also, the drawbacks of the existing methods must be highlighted clearly for justifying the upgrade proposed by the current work. The way the latter work is improving the state of the art must be clarified.

Address: We have rewritten the Introduction section. We added an extra subsection named “1.1 Related works and Motivation” under the Introduction Section on page-2, in line-40. We have rearranged the structure of the manuscript so that it may understandable more.

iii) Novelty of the paper is not mentioned obviously. There're a grammatical and syntax errors.

Address: Thank you for the comments. We have rewritten the Introduction section so that the novelty of this work is focused. All grammatical and syntax errors are revised in a peer review process. The motivation subsection in line-40 may display the novelty of our proposed method. We have added 12 more recent papers and sited them to show the novelty of our work.

iv) Authors are encouraged to introduce a nomenclature section at the beginning of the manuscript, including all variables, acronyms, indexes and constants defined in the manuscript, in order to make the text more clear and readable.

Address: Thank you so much for the comments. A nomenclature in item formatted has been introduced in Section-3 from lines 186-203.

v) In the introduction section, the authors need more to introduce the previous literature. A comprehensive paper needs more than 40 references at least.

Address: We have rewritten the Introduction section and accommodated 12 more papers from previous literatures. We added an extra subsection named “Related works and Motivation” under the Introduction Section (line 39 to 106). We have cited more relevant papers from the recent publications. New references are [24-25], [30-35] and [38-41].

vi) Figures should be given with better accuracy and described in the paper. Figures must be replaced with high resolution ones.

Address: We have simulated all results with better accuracy. Some of them are redrawn with illustrator for better dpi. Figures are replaced with higher resolutions. The redrawn figures are Fig-1, Fig-5, Fig-6, Fig-8, Fig-9, and Fig-10.

vii) Authors must be talk about the future work and potential limitations briefly in the Conclusions and Recommendations section.

Address: Thank you for the comments. We have changed the section name from “Conclusions” to “Conclusion and Recommendations”. We added future scope from lines 463-472.

viii) Variables in the text must be italic.

Address: All variables in the manuscript are changed to italic form.

iX) All those comments are unfortunate when we see the quality of the numerical results. Via those results, the proposed method demonstrates its effectiveness without any doubt. However, the materials for introducing the state of the art and the methodology is deficient and poor.

Address: Thank you for the comment. We have tried our best to reflect the state of the art by rewriting the introduction section, adding the nomenclatures of different variables, redrawing some figures and citing 12 more papers. Fig-1, Fig-8, Fig-10 and Table-3 are updated. We have added Table-4 for a comparison to practical, simulated and existing results.

Reviewer #2: An Approach to Increase the Power Output of Karnafuli Hydroelectric Power Station: A Step to Sustainable Development in Bangladesh’s Energy Sector Although it was felt that your manuscript was quite interesting. The main problems with the paper are as follow:

1-First paragraph of the introduction needs to be rewritten. (You can omit this paragraph)

Address: Thank you so much for the comment. We have omitted this paragraph from the Introduction Section. 

2-“In this work, a method is proposed to obtain additional power from the existing plant utilizing the Combined-Cycle Hydropower System (CCHS) technology to increase overall plant capacity [12].” is it done in reference [12]?!

Address: Thank you for the comment. Mistakenly, we have cited a review paper [12] in the Introduction section in lines 40-43. Sorry for the mistake. We have omitted this reference from this section and can be seen from lines 106-107.

3-Introduction needs to be rewritten. Please review more papers and mention their method weaknesses.

Address: We have rewritten the Introduction section and cited 12 more papers from previous literatures. We added an extra subsection named “Related works and Motivation” under the Introduction Section (line 39 to 106). We have cited more relevant papers from the recent publications and displayed their limitation and scope of improvements. New references are [24-25], [30-35] and [38-41].

4-"An Approach to Increase the Power Output of Karnafuli Hydroelectric Power Station" How much did it increase?

Address: Thank you for the comments. It was mentioned (12.2 MW) in the “Simulation Result” Section 6.2 at line 389, unfortunately, we had a wrong referencing Table-6 for comparison. There was no Table-6 in the manuscript, however it should be Table-3. Sorry for the mistake. Now we add this value in Table-3 and compared the value with the estimated (13.32 MW) from surveyed data from the plant in line 390.

5-The main concern with this paper is the contribution of the paper. The mentioned contributions cannot be new idea.

Address: Thank you for the comment. We agreed with you. However, the rewritten Introduction section may reflect our contribution now. The previous works are based on river natural water current. Our contributions are: 

• Real data from the plant: Survey in Kaptai hydropower plant to collect data. We collect the real data of the KH plant.

• Analysis to data: Primary analysis and power estimation are performed based on surveyed data. A simulation model is established to analyze the existing KH plant.

• Site Selection: There were 4 regions and we have selected the optimum one with proper reasoning. 

• Combined Technology: CCHS technology is old idea, but we apply it in the plant with analysis and site selection process. Based on the analysis, we proposed the suitable turbine to generate 27% more electricity from the plant. 

• A prototype model: To verify the proposed model we built a pilot project in a small scale and compared with the simulated results. 

6-Studies have been performed on the Karnafuli Hydroelectric Power Station. Please compare simulation results with practical results (if possible).

Address: Thank you for the comment. We have added Table-4 in section 6.4 “Discussion and Comparison” showing a comparison among the estimated, simulated and experimental results. Lines 434-437 are added to the section to show the comparison.

Table 4. A comparison among estimated, simulated and experimental results.

Parameters Simulated

(Unit-4 of KH) Estimated

(from Data) Simulated (proposed) Prototype 

Model 

Voltage 100kV 98 kV 7 kV 2.9 V

Current 190A 151 A 1500A 324 mA

Frequency 50 Hz 50 Hz 50 Hz 50 Hz

Average, P 17.1 MW 13.32 MW 12.20 MW 0.78 Watt

-References should be written as standard

Address: All references are rechecked and maintained with journal standard and Plos One standard.

-In addition, proofread the text to avoid fragments, compliance with English grammar and punctuations rules and subject and verb agreement.

Address: Thank you for the comment. We have double checked the manuscript thoroughly.

---

## [Decision Letter · Decision Letter 1]

7 Sep 2021

An Approach to Increase the Power Output of Karnafuli Hydroelectric Power Station: A Step to Sustainable Development in Bangladesh's Energy Sector

PONE-D-21-18420R1

Dear Dr. Hossain,

We’re pleased to inform you that your manuscript has been judged scientifically suitable for publication and will be formally accepted for publication once it meets all outstanding technical requirements.

Kind regards,

Mehrdad Ahmadi Kamarposhti

Academic Editor

PLOS ONE

Additional Editor Comments (optional):

Reviewers' comments:

Reviewer's Responses to Questions

**Comments to the Author**

1. If the authors have adequately addressed your comments raised in a previous round of review and you feel that this manuscript is now acceptable for publication, you may indicate that here to bypass the “Comments to the Author” section, enter your conflict of interest statement in the “Confidential to Editor” section, and submit your "Accept" recommendation.

Reviewer #1: All comments have been addressed

Reviewer #2: All comments have been addressed

2. Is the manuscript technically sound, and do the data support the conclusions?

Reviewer #1: Yes

Reviewer #2: Yes

3. Has the statistical analysis been performed appropriately and rigorously? 

Reviewer #1: Yes

Reviewer #2: Yes

4. Have the authors made all data underlying the findings in their manuscript fully available?

Reviewer #1: Yes

Reviewer #2: Yes

5. Is the manuscript presented in an intelligible fashion and written in standard English?

Reviewer #1: Yes

Reviewer #2: Yes

6. Review Comments to the Author

Reviewer #1: I have no further comments on the document. All comments are well addressed. The paper is now suitable for publication

Reviewer #2: (No Response)

7. PLOS authors have the option to publish the peer review history of their article (what does this mean?). If published, this will include your full peer review and any attached files.

Reviewer #1: No

Reviewer #2: No

---

## [Editor Report · Acceptance letter]

28 Sep 2021

PONE-D-21-18420R1 

An Approach to Increase the Power Output of Karnafuli Hydroelectric Power Station: A Step to Sustainable Development in Bangladesh's Energy Sector 

Dear Dr. Hossain:

I'm pleased to inform you that your manuscript has been deemed suitable for publication in PLOS ONE. Congratulations! Your manuscript is now with our production department. 

Kind regards, 

on behalf of

Dr. Mehrdad Ahmadi Kamarposhti 

Academic Editor

PLOS ONE